# Unveiling the Involvement of Extracellular Vesicles in Breast Cancer’s Organotrophic Metastasis: Molecular Mechanisms and Translational Prospects

**DOI:** 10.3390/ijms26125430

**Published:** 2025-06-06

**Authors:** Haotian Shang, Yumin Zhang, Tengfei Chao

**Affiliations:** Department of Oncology, Tongji Hospital, Tongji Medical College, Huazhong University of Science and Technology, Wuhan 430030, China; m202276295@hust.edu.cn (H.S.); m202476647@hust.edu.cn (Y.Z.)

**Keywords:** extracellular vesicles, breast cancer, metastasis, breast cancer biomarkers

## Abstract

Breast cancer metastasis remains the primary driver of patient mortality, involving dynamic interactions between tumor cells and distant organ microenvironments. In recent years, tumor cell-derived extracellular vesicles (EVs) have emerged as critical information carriers, playing central roles in breast cancer metastasis by mediating organ-specific pre-metastatic niche formation, immune modulation, and tumor cell adaptive evolution. Studies have demonstrated that EVs drive the metastatic cascade through the delivery of bioactive components, including nucleic acids (e.g., miRNAs, circRNAs), proteins (e.g., integrins, metabolic enzymes), and lipids, which collectively regulate osteoclast activation, immune cell polarization, vascular permeability alterations, and extracellular matrix (ECM) remodeling in target organs such as bone, the lungs, and the liver. Molecular heterogeneity in EVs derived from different breast cancer subtypes strongly correlates with organotropism, providing potential biomarkers for metastasis prediction. Leveraging the organotrophic mechanisms of EVs and their dual regulatory roles in metastasis (pro-metastatic and anti-metastatic), strategies targeting EV biogenesis, cargo loading, or delivery exhibits translational potential in diagnostics and therapeutics. In this review, we summarize recent advances in understanding the role of breast cancer-derived exosomes in mediating metastatic organotropism and discuss the potential clinical applications of targeting exosomes as novel diagnostic and therapeutic strategies for breast cancer.

## 1. Introduction

Breast cancer, a highly prevalent malignancy in women, has remained the most frequently diagnosed cancer in women and the leading cause of cancer-related deaths among females since 1980 [1]. In 2022, there were approximately 2.3 million new cases and 670,000 deaths globally [2], making it the second most commonly diagnosed cancer and the fourth leading cause of cancer-related mortality in the same year [3]. In 2022, there were approximately 357,200 newly diagnosed cases of breast cancer among women in China. This makes breast cancer the second most prevalent type among women, with a notably higher incidence rate in urban areas compared to rural regions. Despite significant improvements in prognosis due to advancements in diagnostic technologies and treatment strategies (including endocrine therapy, targeted therapy, immunotherapy, and neoadjuvant therapy) [4,5], patients with distant metastases still face poor clinical outcomes. Therefore, elucidating the mechanisms underlying breast cancer metastasis and developing novel early diagnostic and therapeutic strategies are of critical importance. Numerous studies have revealed the crucial roles of extracellular vesicles (EVs) in breast cancer metastasis. Here, we summarize the roles and underlying mechanisms of tumor cell-derived EVs in regulating breast cancer metastasis, as well as their potential applications in diagnosis and therapy.

### 1.1. Breast Cancer Classification and Metastasis

Breast cancer primarily originates from mammary ducts, lobules, or stromal tissues. It can be classified into several subtypes based on the expression of molecular markers. These markers include Human Epidermal Growth Factor Receptor 2 (HER-2), estrogen receptors (ERs), and progesterone receptors (PRs), which serve as therapeutic targets for breast cancer drugs and indicators for selecting effective treatment strategies. Furthermore, these molecular markers are critical for evaluating the likelihood of metastasis and disease progression in breast cancer patients [6].

Cancer metastasis involves a series of sequential processes, including detachment from the primary tumor, intravasation into blood or lymphatic vessels, survival in the bloodstream as circulating tumor cells (CTCs), extravasation of CTCs from the circulatory system into target organs, and subsequent colonization [3]. Recurrence and metastasis of breast cancer represent the leading causes of mortality [4], with distinct organ-specific tropism observed in metastatic patterns. Bone metastases are the most prevalent, accounting for 50.7–68.8% of all metastatic cases, followed by pulmonary metastases (16.0–23.9%), hepatic metastases (13.3–19.7% of metastatic cases), and, the least frequent, brain metastases (1.9–5.7%) [5,6]. Successful metastatic dissemination requires not only the intrinsic aggressiveness of tumor cells but also the establishment of a permissive microenvironment at secondary sites to facilitate tumor cell survival and proliferation [7]. Current research has demonstrated that exosomes secreted by tumor cells play a pivotal role in orchestrating metastatic progression [8].

### 1.2. EVs and Tumor Metastasis

EVs, which are membrane-bound vesicles secreted by a wide range of cell types, play an essential role as crucial mediators of intercellular communication in both physiological and pathological processes [9]. The significance of EVs lies in their ability to convey vital information and substances between cells, thereby influencing various biological functions and responses. Based on their distinct morphological characteristics, EVs are mainly classified into three main classes: exosomes (ranging from 30 to 150 nm), micro vesicles (measuring 150 to 1000 nm), and apoptotic bodies (spanning 500 to 2000 nm) [10]. This classification system provides a framework for understanding the diverse properties and functions of these vesicles.

Emerging research has uncovered that tumor-derived exosomes have highly significant roles in the progression of cancer. Specifically, they exert their influence particularly through modulating the complex metastatic cascades and intricate drug resistance mechanisms that have a profound impact on the course and evolution of the disease [11]. The involvement of tumor-derived exosomes in these processes adds a new layer of complexity to our understanding of cancer development and treatment. These nano-scale vesicles establish intricate molecular communication networks between the secreting cells and the recipient cells by transporting a diverse array of cargoes. These cargoes encompass proteins with various functions and structures, nucleic acids in different forms such as DNA and RNA, and lipids of diverse compositions [12,13,14]. The transportation of these cargoes enables the exchange of essential molecules and signals, facilitating intercellular communication and regulating various cellular processes.

Exosomes contribute to organotrophic regulation through multiple pathways, encompassing vascular permeability modulation, induction of epithelial–mesenchymal transition (EMT), facilitation of immune evasion, and formation of PMNs [11,15,16]. Proteomic analyses have demonstrated that exosomes secreted by different breast cancer subtypes exhibit distinct protein profiles, with these molecular disparities showing significant correlations with organ-specific metastasis patterns, thereby providing crucial insights into the molecular underpinnings of tumor dissemination [17].

In this review, we will systematically elucidate the subtype-specific regulatory mechanisms of exosomes in breast cancer bone, pulmonary, hepatic, and brain metastases, accompanied by an in-depth exploration of their targeted delivery mechanisms.

## 2. EVs and Organ-Specific Metastasis of Breast Cancer

### 2.1. Bone Metastasis of Breast Cancer

Bone metastasis, as the most prevalent metastatic manifestation of breast cancer, typically induces skeletal morbidity, including pathological fractures, spinal cord compression, and hypercalcemia [18,19,20]. This metastatic process reduces overall survival rates while causing mobility impairment and loss of social functioning, diminished quality of life, and increased healthcare expenditures. Radiologically classified by lesion characteristics, bone metastases are categorized into osteolytic (bone-destructive) and osteoblastic (bone-forming) subtypes. Osteolytic metastases manifest through focal bone destruction dominated by osteoclast-mediated bone resorption. Conversely, osteoblastic lesions develop when enhanced osteoblast activity leads to sclerotic bone formation [21]. Breast cancer bone metastases predominantly exhibit osteolytic characteristics, primarily mediated through IL-11 secretion, which stimulates osteoclastogenesis and bone destruction, MMP1-driven tumor cell invasion, and subsequent amplification via the RANK/RANKL signaling cascade [22,23]. Furthermore, additional factors including connective tissue growth factor (CTGF), CXCR4 chemokine receptor, and various other molecular mediators significantly influence the progression of breast cancer bone metastases [24].

Recent studies have EVs play crucial roles in breast cancer bone metastasis, primarily through miRNA-mediated enhancement of osteoclast function. These EVs facilitate the transportation of specific miRNAs to bone tissue, thereby exacerbating osteolytic bone destruction (Figure 1). For instance, miR-218 secreted by breast cancer cells via exosomes directly targets COL1A1 in osteoblasts, suppressing type I collagen expression and secretion to shift the bone remodeling balance toward resorption. Additionally, miR-218 upregulates inhibin beta A subunit (INHBA) expression and secretion by targeting both YY1 and inhibin beta B subunit (INHBB). Secreted INHBA inhibits SMAD2/3 signaling in osteoblasts, inducing Timp3 expression to impair type I collagen processing, reduce bone formation, and create a bone microenvironment favorable for breast cancer progression [25]. Breast cancer cells also secrete exosomes loaded with miR-940, which enhance the osteogenic differentiation potential of mesenchymal stem cells (MSCs) by suppressing the expression of Rho GTPase-activating protein 1 (ARHGAP1), thereby inhibiting the activation of the RhoA/ROCK signaling pathway. Additionally, miR-940 suppresses the expression of FAM134A, triggering the unfolded protein response (UPR) in the endoplasmic reticulum. This activates the IRE1α-XBP1 pathway, leading to the upregulation of RUNX2 expression, which in turn activates osteogenic differentiation-related signaling pathways in MSCs and promotes their osteogenic differentiation. This process is characterized by increased bone matrix mineralization and elevated expression of osteogenic markers [26]. Notably, studies have revealed that highly metastatic SCP28 breast cancer cells secrete exosomes encapsulating miR-21, which are delivered to osteoclast precursors. These exosomes suppress the expression of programmed cell death 4 (PDCD4), thereby alleviating its inhibitory effect on the transcription factor NFATc1. This mechanism activates key pathways for osteoclast differentiation (e.g., upregulation of markers such as TRAP, CTSK and MMP9), promoting the formation of an osteolytic microenvironment. This exosome-mediated bone microenvironment remodeling is characterized by reduced bone density and enhanced osteoclast activity, creating favorable conditions for tumor cell colonization. Clinical analyses demonstrate that serum exosomal miR-21 levels are significantly higher in patients with bone metastasis compared to non-metastatic patients, confirming the clinical relevance of this mechanism [27]. Furthermore, highly metastatic triple-negative breast cancer (TNBC) cells MDA-MB-231 secrete exosomes enriched with miR-20a-5p, which are delivered to bone marrow-derived macrophages (BMMs). miR-20a-5p directly targets and binds to the 3′UTR region of the SRCIN1 gene, suppressing the expression of this tumor suppressor. The downregulation of SRCIN1 alleviates its inhibitory effects on downstream signaling pathways (such as the Src kinase pathway), thereby activating key transcription factors for osteoclast differentiation, including NFATc1. This promotes G1/S phase transition in the cell cycle of BMMs, upregulates osteoclast marker genes (e.g., TRAP and cathepsin K), and ultimately drives abnormal proliferation and differentiation of osteoclasts. This exosome-mediated cross-boundary miRNA regulation leads to enhanced bone resorption, releasing growth factors stored in the bone matrix and forming a “tumor cell-osteoclast-bone degradation” positive feedback loop, which accelerates osteolytic bone destruction and breast cancer bone metastasis progression [28]. ER-positive breast cancer cells secrete exosomal miR-19a, which enhances osteoclast activity and bone metastasis through PTEN suppression-mediated activation of NF-κB and AKT pathways [29]. Interestingly, not all miRNAs exhibit pro-osteoclastic effects. miR-24-2-5p demonstrates dual inhibitory functions: it reduces osteoclast differentiation by targeting IL6ST, PEX10, and CNNM4 to maintain bone homeostasis while simultaneously suppressing breast cancer cell migration through inhibition of DCTD, FMR1, and PIGS expression, collectively impairing bone metastasis [30]. Beyond miRNAs, breast cancer-derived exosomes deliver lncRNA-MIR193BHG to osteoclast precursors, where it functions as a competitive endogenous RNA (ceRNA) by sequestering miR-489-3p. This interaction prevents miR-489-3p-mediated suppression of DNMT3A mRNA, leading to DNMT3A upregulation, enhanced IRF8 promoter methylation, and subsequent inhibition of NFATC1 transcriptional activity—a regulatory cascade that ultimately promotes osteoclast differentiation and bone metastasis [31].

Proteins also serve as critical exosomal cargo facilitating bone metastasis progression. TNBC cells secrete exosomes enriched with intercellular cell adhesion molecule-1 (ICAM1), which binds to leukocyte function-associated antigen-1 (LFA-1) on CD8+ T cell surfaces, thereby suppressing CD8+ T cell proliferation and activation, inducing T cell exhaustion, and ultimately promoting TNBC bone metastasis [32]. Beyond secreting miR-19a, ER-positive breast cancer cells also release integrin-binding sialoprotein (IBSP). This protein enhances osteoclast precursor migration and maturation through binding to αVβ3 integrin receptors, synergistically cooperating with miR-19a to amplify tumor cell adaptability within the bone microenvironment and potentiate osteolytic bone metastasis [30]. Furthermore, L-plastin—a member of the actin-binding protein family—released by MDA-MB-231 cells activates calcium signaling and induces nuclear translocation of the NFATC1 transcription factor to drive osteoclastogenesis. Notably, L-plastin cooperates with secreted peroxiredoxin-4 (PRDX4) to exacerbate breast cancer-induced osteolysis, collectively facilitating skeletal metastasis progression [33].

### 2.2. Lung Metastasis of Breast Cancer

The lung, as the second most common metastatic site after bone in breast cancer, confers a highly lethal prognosis. Following entry into the bloodstream, breast cancer cells primarily encounter the pulmonary capillary beds. During pulmonary circulation, tumor cells arrest within these capillary beds, subsequently extravasating into lung tissue and developing into metastatic lesions [34,35]. Throughout this process, exosomes play crucial roles in enhancing cancer cell invasiveness, remodeling the immune microenvironment, and facilitating PMN formation (Figure 2).

Macrophages, as pivotal components of the immune microenvironment, exert critical regulatory roles in cancer metastasis. Breast cancer cell-derived exosomes encapsulate miR-138-5p, which is transferred via paracrine signaling to tumor-associated macrophages (TAMs). Within TAMs, miR-138-5p directly targets the mRNA of *KDM6B*, suppressing its expression and consequently elevating H3K27me3 modification levels. This epigenetic alteration drives M2 macrophage polarization. The polarized M2 macrophages secrete vascular endothelial growth factor (VEGFA) and immunosuppressive factors such as interleukin-10 (IL-10), thereby remodeling the pulmonary PMN and facilitating breast cancer cell colonization in the lung [36]. Beyond miRNAs, circular RNAs (circRNAs) also modulate macrophage functionality to promote lung metastasis. For instance, circ-0100519 is packaged into exosomes and internalized by macrophages. Functioning as a scaffold protein, circ-0100519 enhances the interaction between ubiquitin-specific protease 7 (USP7) and nuclear factor erythroid 2-related factor 2 (NRF2), thereby inhibiting NRF2 ubiquitination and degradation. This stabilization activates pro-inflammatory factors including IL-10 and transforming growth factor-beta (TGF-β), which drive M2 macrophage polarization. The M2 macrophages subsequently secrete immunosuppressive molecules (e.g., arginase 1 [Arg1], CD206) and metastasis-promoting factors such as matrix metalloproteinase 9 (MMP9), collectively accelerating breast cancer lung metastasis [37]. Proteins transported via exosomes further regulate macrophage activity. Signal-induced proliferation-associated protein 1 (SIPA1) promotes the secretion of exosomes enriched with myosin heavy chain 9 (MYH9) from breast cancer cells. Upon transfer to macrophages, MYH9 stimulates the secretion of MMP2 and VEGFA, which promote angiogenesis, ECM degradation, and the establishment of a pro-metastatic lung microenvironment [38]. Under hypoxic conditions, breast cancer cells secrete autophagy marker LC3-enriched exosomes, termed tumor-derived autophagy-related particles (TRAPs). Surface heat shock protein 60 (HSP60) on TRAPs binds to Toll-like receptor 2 (TLR2) on lung fibroblasts, activating the myeloid differentiation primary response 88 (MyD88)-nuclear factor-kappa B (NF-κB) pathway. This cascade induces chemokine (C-C motif) ligand 2 (CCL2) secretion, which recruits monocytes differentiating into M2 macrophages while concurrently suppressing CD8+ T cell function, fostering an immunosuppressive microenvironment. Simultaneously, fibroblast-derived TGF-β promotes EMT in tumor cells, enhancing their invasive capacity and facilitating pulmonary metastasis [39]. Neutrophils, indispensable constituents of the immune landscape, also critically influence metastatic progression. In breast cancer cells with elevated Lin28B expression, the maturation of let-7 miRNAs is disrupted. Lung neutrophils internalizing exosomes containing low let-7 levels exhibit activation of the signal transducer and activator of transcription 3 (STAT3) pathway, driving their polarization into pro-tumorigenic N2 neutrophils. These N2 neutrophils secrete S100 calcium-binding protein A8/A9 (S100A8/A9), which orchestrates PMN formation in the lung [40]. Exosomes encapsulating transcription factor SP1 are internalized by pulmonary neutrophils, activating the TLR4-NF-κB pathway to induce CXCL2 secretion. This establishes a neutrophil autocrine loop, wherein reactive oxygen species (ROS) and VEGFA increase vascular permeability, enhance tumor cell extravasation, and suppress CD8+ T cell activity, collectively shaping an immunosuppressive niche conducive to metastasis [41]. Breast cancer cell-derived caveolin-1 (CAV1)-positive exosomes, enriched with integrin α6β4 on their surface, activate lung neutrophils via the TLR4-NF-κB pathway, driving their polarization into the N2 phenotype. The polarized neutrophils release MMP9 and S100A8/A9, which degrade the pulmonary basement membrane and promote tumor cell colonization [42]. Notably, integrin α6β4 also interacts with surfactant-associated protein C (SFTPC) on lung epithelial cells, enhancing tumor cell tropism to the lung and facilitating metastasis [42]. The metastatic potential of cancer cells is intrinsically linked to their invasive capacity. Loss of DENND10, a regulator of vesicular trafficking, reduces the abundance of ECM components such as fibronectin and laminin within breast cancer exosomes. This impairs tumor cell adhesion to lung tissue and attenuates migratory and invasive capabilities [43]. The long non-coding RNA NEAT1, secreted via exosomes, functions as a ceRNA by sponging miR-141-3p. This interaction derepresses Kruppel-like factor 12 (KLF12), activating the EMT, upregulating MMPs and ATP-binding cassette (ABC) transporters, and ultimately enhancing tumor cell invasion and metastasis [44]. The integrity of pulmonary vasculature profoundly impacts metastatic efficiency. Breast cancer exosomes carrying nucleoside diphosphate kinase B (NDPK-B) catalyze the conversion of adenosine diphosphate (ADP) to adenosine triphosphate (ATP). ATP activates purinergic receptor P2Y1 on lung endothelial cells, inducing vascular endothelial growth factor receptor 2 (VEGFR2) phosphorylation. This cascade promotes vascular neoangiogenesis and leakage, facilitating circulating tumor cell (CTC) extravasation and metastatic lesion formation [45].

### 2.3. Liver Metastasis of Breast Cancer

The liver represents a common metastatic site for all solid cancers and ranks as the third most frequent metastatic location in breast cancer. Distinct breast cancer subtypes exhibit differential organotropism in metastatic dissemination, with HER-2-positive tumors demonstrating significantly elevated hepatic metastasis risk [5,6]. During hepatic metastasis progression in breast cancer, multiple factors, including the inflammatory cytokine TNF-α, the chemokine/receptor pair CXCL12/CXCR4, and cell adhesion molecules such as cadherins, collectively influence metastatic advancement [46,47,48].

Emerging research has elucidated the critical role of extracellular vesicle (EV)-mediated miRNA signaling in regulating breast cancer hepatic metastasis (Figure 3). Breast cancer stem cells (BCSCs) secrete EVs that deliver miR-197 to suppress PPARG expression via targeting its 3′UTR region, thereby inducing the EMT. Mechanistically, miR-197-mediated PPARG downregulation reduces E-cadherin levels while upregulating mesenchymal markers like Vimentin, collectively enhancing tumor cell invasiveness and hepatic metastasis [49]. Furthermore, miRNAs can modulate hepatic metastasis through cholesterol metabolism regulation. Breast cancer-derived EVs transport miR-9-5p to upregulate hepatic cholesterol synthesis enzymes (HMGCR, DHCR7) and oxidase CH25H, resulting in 25-hydroxycholesterol (25-HC) accumulation. This metabolite activates LXR signaling to enhance tumor cell stemness while inducing M2 macrophage polarization via the IL-6/STAT3 pathway, creating an immunosuppressive microenvironment conducive to metastatic progression [50]. TNBC, the most aggressive subtype, secretes EVs that stimulate hepatic endothelial cells to release CX3CL1, recruiting CX3CR1+ macrophages to PMNs. These macrophages overexpress MMP9 to promote vascular leakage and tumor cell extravasation, effectively reducing hepatic metastatic resistance [51]. Additionally, Tspan8—a tetraspanin family member—enhances intercellular adhesion via E-cadherin/β-catenin signaling while augmenting EV secretion. Tspan8+ EVs carry E-cadherin and p120-catenin to induce the mesenchymal–epithelial transition (MET) in metastatic tumor cells, thereby strengthening their colonization capacity [52].

### 2.4. Brain Metastasis of Breast Cancer

Brain metastasis represents one of the principal causes of mortality in breast cancer patients, occurring in approximately 30% of metastatic breast cancer cases, with a median survival duration of merely 6–18 months [53,54]. The establishment of brain metastases involves intricate processes including tumor cell traversal across the blood–brain barrier (BBB), adaptation to the neuro-metabolic microenvironment, and immune surveillance evasion [55]. Recent advances have demonstrated that breast cancer-derived EVs play pivotal roles in remodeling the cerebral microenvironment through the delivery of functional molecules (miRNAs, proteins, etc.), serving as central mediators in the “seed–soil” interactions underlying cerebral metastatic progression (Figure 4).

EVs critically regulate BBB permeability and tumor cell invasion during cerebral metastasis. Studies demonstrate that brain-tropic breast cancer EVs (Br-EVs) enhance BBB permeability by downregulating endothelial Rab11fip2 (a regulator of vesicle recycling) while upregulating Rab11fip3/5, thereby inducing cytoskeletal reorganization in brain microvascular endothelial cells [56]. Br-EVs carrying the sodium–potassium–chloride cotransporter NKCC1 further compromise barrier integrity by increasing endothelial cell volume and motility [56]. Cerebral metastatic cells utilize CEMIP (cell migration-inducing and hyaluronan-binding protein)-enriched EVs to promote tumor-endothelial adhesion. CEMIP+ EVs activate endothelial branching and induce pro-inflammatory factors (Ptgs2, Tnf, etc.), establishing a pro-metastatic vascular niche. Genetic ablation of CEMIP significantly impairs tumor-vascular co-option capacity [57]. Breast cancer cells with brain metastasis tropism secrete EVs that are internalized by BBB astrocytes via the Cdc42-dependent CLIC/GEEC endocytic pathway. Br-EVs surface-enriched proteins, including CD73, uPAR, and integrin β1, interact with GPI-anchored proteins on astrocyte membranes to promote their endocytosis. These EVs deliver miR-301a-3p to astrocytes, specifically targeting the 3′UTR of TIMP-2 to suppress its expression, thereby reducing levels of the matrix metalloproteinase inhibitor TIMP-2 and triggering ECM remodeling. In vivo experiments confirm that Br-EVs traverse the BBB intact through transcytosis, significantly reducing cerebral TIMP-2 levels (more pronounced than primary tumor-derived EVs), while miR-301a-3p levels show an inverse correlation with TIMP-2. This matrix regulatory effect does not compromise BBB integrity but creates a pro-invasive microenvironment conducive to metastatic niche formation [58]. Astrocytes play pivotal roles in brain metastasis development. Cerebral metastasis-derived EVs overexpressing miR-199b-5p target SLC1A2 (glutamate transporter) in astrocytes and SLC38A2/SLC16A7 (glutamine/lactate transporters) in neurons, resulting in an extracellular accumulation of glutamate, glutamine, and lactate, which fuels metastatic cell metabolism [59]. Furthermore, EVs deliver miR-1290/1246 to suppress FOXA2 transcription factor, thereby relieving its inhibition on ciliary neurotrophic factor (CNTF). This mechanism induces the M2 polarization of astrocytes, driving the secretion of pro-inflammatory cytokines (IL-6, VEGF, etc.) and promoting cancer stem cell expansion [60]. Breast cancer-derived EVs also reprogram the immune microenvironment to facilitate cerebral metastasis. XIST-deficient breast cancer EVs suppress TIMP-2 via exosomal miR-503, polarizing microglia toward an M2 phenotype that secretes immunosuppressive factors (IL-10, TGF-β) to inhibit T-cell proliferation. Concurrently, these EVs activate the MSN-c-Met pathway to enhance tumor cell stemness and the EMT [61].

## 3. Clinical Diagnostics and Applications

Early diagnosis of breast cancer exerts a decisive impact on patient survival outcomes. Current research advancements highlight the emerging potential of tumor-derived exosomes as liquid biopsy biomarkers which can be isolated and analyzed from various biofluids, including blood, urine, breast milk, saliva, ascites, and cerebrospinal fluid [62,63,64,65,66,67]. These nanosized vesicles carry tumor-specific molecular signatures that dynamically reflect disease staging and subtype heterogeneity, thereby providing revolutionary opportunities for developing next-generation diagnostic platforms. The intrinsic advantages of exosomal biomarkers—including their abundance in bodily fluids, molecular stability, and real-time monitoring capability—position them as superior alternatives to traditional tissue biopsies for longitudinal disease surveillance. Proteomic profiling has identified exosome-associated proteins with diagnostic and prognostic relevance. Lee et al. conducted comprehensive LC-MS/MS analysis of small extracellular vesicles (sEVs) derived from aggressive breast cancer cell lines, identifying 270 sEV-associated proteins. Through functional validation, EGF-like repeats and discoidin I-like domains-containing protein 3 (EDIL3) emerged as a metastasis-correlated diagnostic biomarker, with its expression levels demonstrating significant associations with clinical progression parameters [68]. In parallel, Risha et al. performed nano LC-MS/MS-based proteomic characterization of TNBC cells, revealing 726 uniquely expressed proteins. Comparative analysis identified three membrane surface markers—glypican-1 (GPC-1), glucose transporter-1 (GLUT-1), and ADAM metallopeptidase domain 10 (ADAM10)—that exhibited substantial upregulation in TNBC-derived sEVs compared to non-malignant controls. These membrane-anchored proteins demonstrate particular clinical relevance as they enable antibody-based detection strategies without vesicle lysis, highlighting their potential for point-of-care diagnostic applications [69]. These systematic investigations collectively underscore the capacity of exosomal proteomic signatures to stratify tumor biological behavior while providing molecular insights for precision diagnostics.

Exosomes, leveraging their inherent delivery capabilities to traverse biological barriers like the BBB while protecting encapsulated cargo from enzymatic degradation and immune clearance, have emerged as promising therapeutic vectors for cancer treatment [70]. Their natural phospholipid bilayer structure enables an efficient encapsulation of diverse bioactive molecules, including chemotherapeutic agents, nucleic acid therapeutics (siRNA, miRNA, mRNA), and immunomodulators, thereby minimizing the systemic toxicity associated with free drug administration. Strategic bioengineering approaches—such as surface conjugation of antibody fragments for antigen-specific targeting, glycosylation modifications to enhance tissue tropism, and pH-responsive membrane modifications for controlled release—have significantly advanced their precision delivery potential.

A notable therapeutic application involves engineered exosomes loaded with S100A4 siRNA (siS100A4), designed to specifically target the PMN in lungs. This strategy demonstrated remarkable suppression of postoperative metastasis in TNBC models by silencing S100A4-mediated pro-metastatic signaling pathways [71]. Furthermore, exosomes purified from mesothelin (MSLN)-targeted chimeric antigen receptor (CAR) T cells exhibit superior therapeutic efficacy compared to parental CAR-T cells. These CAR-T-derived exosomes effectively inhibit MSLN-positive breast cancer cell proliferation while circumventing the cytokine release syndrome and neurotoxicity risks inherent in conventional CAR-T therapies, showcasing their potential as safer and more potent alternatives in adoptive cell therapy [72,73]. The intrinsic advantages of exosomes, including low immunogenicity, enhanced tumor penetration, and multi-cargo loading capacity, position them as transformative tools in next-generation precision oncology.

## 4. Conclusions

Current research has thoroughly elucidated exosome-mediated regulation of breast cancer metastasis through diverse pathways and cellular targets, highlighting their potential as therapeutic intervention points (Table 1). Modulating exosome secretion or engineering their cargo composition emerges as a viable strategy to suppress metastatic progression. To advance clinical translation, critical priorities include developing standardized methods to differentiate pathogenic from therapeutic exosome subsets, establishing rigorous purification and analytical protocols, and validating subtype-specific biomarkers.

These advancements will enable precise interception of metastasis-promoting exosomes while harnessing engineered exosomes as targeted therapeutic vectors, ultimately paving the way for novel diagnostic and treatment paradigms.

## Figures and Tables

**Figure 1 ijms-26-05430-f001:**
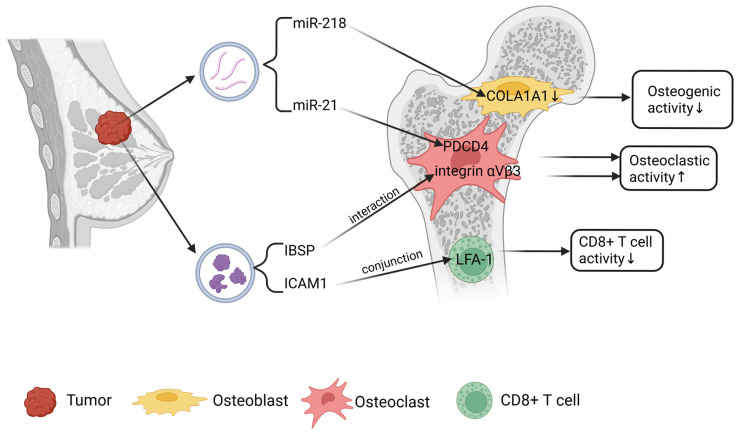
Mechanisms of EVs mediating bone metastasis of breast cancer. Figure created in BioRender. Chao, T. (2025) https://BioRender.com/h42n006.

**Figure 2 ijms-26-05430-f002:**
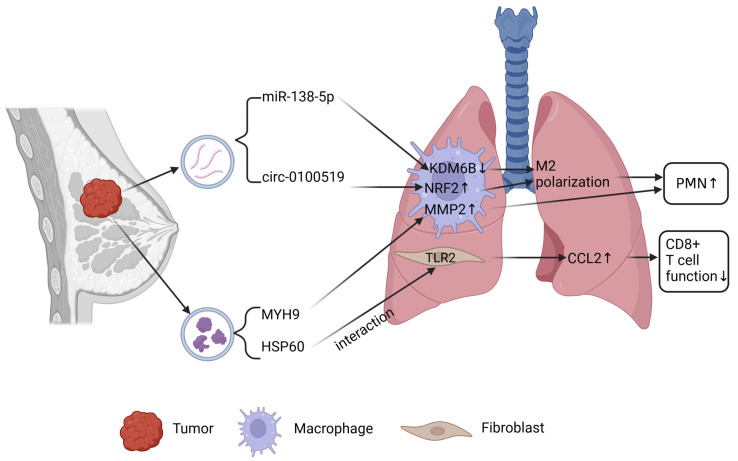
Mechanisms of EVs mediating lung metastasis of breast cancer. Figure created in BioRender. Chao, T. (2025) https://BioRender.com/h42n006.

**Figure 3 ijms-26-05430-f003:**
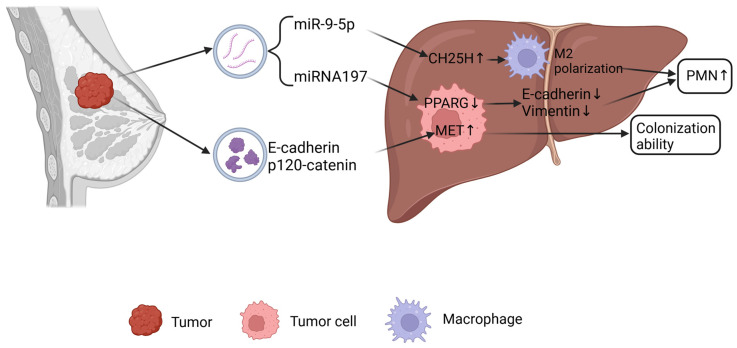
Mechanisms of EVs mediating lung metastasis of liver cancer. Figure created in BioRender. Chao, T. (2025) https://BioRender.com/h42n006.

**Figure 4 ijms-26-05430-f004:**
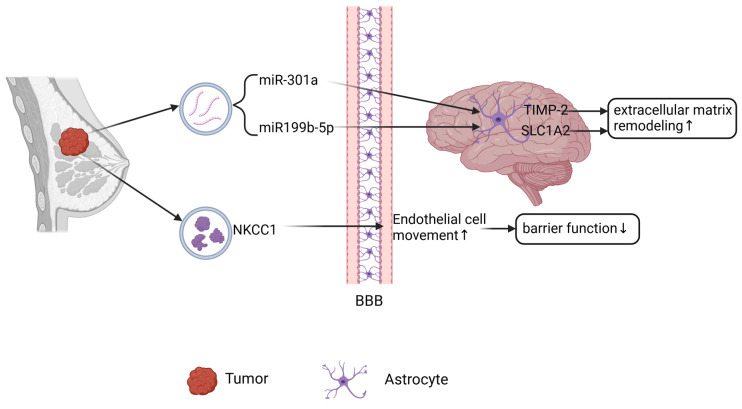
Mechanisms of EVs mediating brain metastasis of breast cancer. Figure created in BioRender. Chao, T. (2025) https://BioRender.com/h42n006.

**Table 1 ijms-26-05430-t001:** Summary of the effects of different EVs on tumor metastasis to different organs.

Metastatic Site	EV-Carried Molecules	Mechanisms	Function
Bone	miR-218	Inhibits COL1A1 expression, promotes bone resorption	Promote
	miR-940	Enhances osteogenic differentiation of MSCs	Promote
	miR-21	Activates osteoclast differentiation pathways	Promote
	miR-20a-5p	Suppresses SRCIN1, promotes osteoclast proliferation	Promote
	miR-19a	Inhibits PTEN, activates NF-κB/AKT pathways	Promote
	ICAM1	Induces CD8+ T cell exhaustion	Promote
	IBSP	Enhances migration of osteoclast precursors	Promote
	L-plastin	Activates calcium signaling and NFATC1	Promote
	miR-24-2-5p	Inhibits osteoclast differentiation and tumor migration	Inhibit
Lung	miR-138-5p	Induces M2 macrophage polarization | Promote	Promote
	circ-0100519	Stabilizes NRF2, promotes pro-inflammatory factors	Promote
	MYH9	Stimulates MMP2/VEGFA secretion | Promote	Promote
	LC3+ EVs (TRAPs)	Activates TLR2-MyD88-NF-κB pathway	Promote
	Low let-7 EVs	Activates STAT3, polarizes N2 neutrophils	Promote
	SP1+ EVs	Activates TLR4-NF-κB pathway | Promote	Promote
	CAV1+ EVs	Promotes N2 neutrophil polarization	Promote
	DENND10-deficient EVs	Reduces ECM components, impairs cell adhesion	Inhibit
	NEAT1	Activates EMT and invasion-related genes | Promote	Promote
Liver	miR-197	Suppresses PPARG, induces EMT | Promote	Promote
	miR-9-5p	Modulates cholesterol metabolism	Promote
	CX3CL1-inducing EVs	Recruits CX3CR1+ macrophages	Promote
	Tspan8+ EVs	Induces MET to enhance colonization	Promote
Brain	CEMIP+ EVs	Promotes tumor endothelial adhesion	Promote
	miR-301a-3p	Suppresses TIMP-2, remodels ECM	Promote
	miR-199b-5p	Alters neuro-metabolic microenvironment	Promote
	miR-1290/1246	Induces M2 astrocyte polarization	Promote
	XIST-deficient EVs	Suppresses TIMP-2, activates immunosuppression	Promote

## Data Availability

Not applicable.

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
