# Peer review of "Unveiling the Involvement of Extracellular Vesicles in Breast Cancer’s Organotrophic Metastasis: Molecular Mechanisms and Translational Prospects"

_ijms, 2025, doi:10.3390/ijms26125430_

Round 1
Reviewer 1 Report
Comments and Suggestions for Authors
Review 1
Unveiling the Involvement of Extracellular Vesicles in Breast 2
Cancer's Organotrophic Metastasis: Molecular Mechanisms and 3
Translational Prospects
The manuscript describes the role of vesicles in diagnostics and clinical potential for use in the treatment of breast cancer. Overall, the text is coherent, has an appropriate amount of literature and precise figures that well illustrate the issues described.
One sentence requires correction (lines 345-347):
Current research advancements highlight the emerging potential of tumor-derived exosomes as liquid biopsy biomarkers, which can be non-invasively isolated and analyzed from various biofluids including blood, urine, breast milk, saliva, ascites, and cerebrospinal fluid
it is difficult to say that collecting cerebrospinal fluid is a non-invasive procedure. Generally, collecting urine, breast milk and saliva is non-invasive, blood sampling is minimally invasive, and cerebrospinal fluid is a procedure that carries some risk. These issues require explanation. In summary, it is good to indicate what new aspects this text brings - there are already many works on vesicles in cancers, including breast cancer. The authors did not indicate a single implemented application of knowledge about vesicles in clinical practice. All the issues discussed have potential applications. Are there really no clinical applications, e.g. in diagnostics or monitoring the course of the disease, of marking such vesicles? These issues should be explained.
Author Response
This sentence has been modified and marked in red.
We have extensively reviewed the literature and investigated ongoing clinical trials, but unfortunately, we have not identified any practical applications of EVs in clinical diagnosis or therapeutic interventions.
Reviewer 2 Report
Comments and Suggestions for Authors
The authors reviewed the roles of extracellular vesicles (EVs) and the underlying mechanisms involved in breast cancer metastasis. EVs serve as carriers of molecular information, including miRNAs and proteins, and play a significant role in the metastatic progression of breast cancer.
This work makes a valuable contribution to advancing our understanding of the importance of EVs in breast cancer metastasis.
Author Response
Thank you for your thorough review of our manuscript and your valuable recognition of our work. We fully endorse the central thesis you highlighted regarding "the pivotal role of extracellular vesicles (EVs) as molecular information carriers in breast cancer metastasis." Through our systematic elucidation of EV-mediated organ-specific metastatic mechanisms—including miRNA/protein delivery, metabolic reprogramming, and immune microenvironment modulation—this study aims to offer novel perspectives into the biology of breast cancer metastasis and establish a theoretical foundation for developing EV-based diagnostic and therapeutic strategies.
Reviewer 3 Report
Comments and Suggestions for Authors
Dear Author, I give you the following comment. Please address this in your manuscript to enhance the readability and understanding of your manuscript.
Major Comments – Questions
- How does the inclusion of a schematic figure enhance the understanding of the novel mechanisms proposed in your review?
- What unique aspects of EV-mediated organotropic metastasis are highlighted in your work that differ from previous reviews?
- Can you clearly identify and illustrate the dual roles (pro-metastatic and anti-metastatic) of EVs in a visual model?
- How do breast cancer subtype-specific EV signatures contribute to organotropism, and can this be effectively captured in a schematic?
- In what ways does the proposed schematic figure help clarify the translational implications of targeting EVs in diagnostics and therapeutics?
Minor Comments – Questions
- Could you add a graphical abstract or schematic in the introduction to better communicate the key conceptual framework of the review?
- Is the figure intended to summarize all mechanisms discussed or focus on one organotropic route (e.g., bone vs. lung vs. liver)?
- Have you considered using color coding or labels to distinguish different cargo types (miRNAs, proteins, lipids) in the schematic?
- Would it be useful to visually map the sequence of EV involvement from biogenesis to pre-metastatic niche formation?
- Is there a need to visually differentiate EV roles in immune modulation versus extracellular matrix remodeling in your figure?
These questions aim to address both overarching concerns and specific technical details that could impact the robustness and clarity of the study's findings.
Best Regards
Author Response
Thank you very much for taking the time to review this manuscript. Please find the detailed responses below and the corresponding revisions/corrections highlighted/in track changes in the re-submitted files Question: How does the inclusion of a schematic figure enhance the understanding of the novel mechanisms proposed in your review? Answer: Integrate the EVS-carrying molecules (e.g., miRNAs, circRNAs, and proteins) described separately in the text with their corresponding organ-specific microenvironment remodeling events (e.g., osteoclastic activation and M2 macrophage polarization) into an interaction network. For instance, the involvement of the miR-21→PDCD4→NFATc1 pathway in bone metastasis and the circ-0100519→USP7/NRF2→IL-10 pathway in lung metastasis elucidates the organ-specific regulatory mechanisms. Question: What unique aspects of EV-mediated organotropic metastasis are highlighted in your work that differ from previous reviews? Answer: The novelty of this study lies in our incorporation of emerging metastatic mechanisms, such as the regulatory effects of miR-24-2-5p on breast cancer bone metastasis and miR-9-5p on liver metastasis, among other newly characterized molecular pathways. Question: Can you clearly identify and illustrate the dual roles (pro-metastatic and anti-metastatic) of EVs in a visual model? Answer: I can briefly describe the dual functions of EVs in the form of words: miR-21, miR-20a-5p,ICAM1 IBSP promote bone metastasis. miR-24-2-5P inhibits bone metastasis. miR-138-5p,circ-0100519,MYH9 promotes lung metastasis. siS100A4 inhibits lung metastasis. miR-9-5p promotes liver metastasis. miR-503 and TIMP-2 inhibit brain metastases Question: How do breast cancer subtype-specific EV signatures contribute to organotropism, and can this be effectively captured in a schematic? Answer: In the review, we place greater emphasis on the influence of the diverse cargo carried by EVs on metastasis rather than focusing on the impact of EVs derived from different breast cancer subtypes on metastatic processes. Question: In what ways does the proposed schematic figure help clarify the translational implications of targeting EVs in diagnostics and therapeutics? Answer: We could develop targeted therapeutics based on the cargo contents carried by the EVs depicted in the diagram (such as specific miRNAs, proteins, or metabolites) to facilitate translational therapeutic applications. For instance: miRNA inhibitors: Design antisense oligonucleotides against metastasis-promoting miRNAs like miR-21 or miR-19a Protein blockers: Develop neutralizing antibodies against EV surface integrins (α6β4) or immune-modulatory proteins (ICAM1) Cargo editing: Engineer EVs to deliver therapeutic miRNAs (e.g., miR-24-2-5p mimics) or CRISPR systems to disrupt oncogenic pathways in pre-metastatic niches This approach leverages EV biological specificity while counteracting their pathological roles in organotropic metastasis. Question: Could you add a graphical abstract or schematic in the introduction to better communicate the key conceptual framework of the review? Answer: The framework of this review is organized around the organ-specific metastatic mechanisms mediated by breast cancer-derived EVs. Accordingly, we have presented four distinct schematic diagrams corresponding to bone, lung, liver, and brain metastases, without including additional figures in the Introduction section. Answer: Is the figure intended to summarize all mechanisms discussed or focus on one organotropic route (e.g., bone vs. lung vs. liver)? Question: The schematic aims to comprehensively illustrate all discussed organotropic pathways (bone, lung, liver, and brain metastases) rather than focusing on a single organ alone. Question: Have you considered using color coding or labels to distinguish different cargo types (miRNAs, proteins, lipids) in the schematic? Answer: In the figure, the purple elongated shapes represent miRNAs, while the purple clumps represent proteins. Question: Would it be useful to visually map the sequence of EV involvement from biogenesis to pre-metastatic niche formation? Answer: The figures in the article primarily illustrate the mechanisms by which EVs influence breast cancer metastasis, rather than depicting the biogenesis or formation process of EVs themselves, hence their omission in our illustrations. Question:Is there a need to visually differentiate EV roles in immune modulation versus extracellular matrix remodeling in your figure? Answer: We have already annotated immune regulation or extracellular matrix remodeling in the figures, so no color differentiation was used.